# Feasibility and Early Outcomes of a Parent Training Intervention to Engage Parents in Children’s Media Education

**DOI:** 10.3390/healthcare11152130

**Published:** 2023-07-26

**Authors:** Maria Clara Cavallini, Simona Carla Silvia Caravita, Hildegunn Fandrem

**Affiliations:** 1Centro Tice, 29121 Piacenza, Italy; 2Norwegian Centre for Learning Environment, University of Stavanger, P.O. Box 8600, 4036 Stavanger, Norway; simona.c.caravita@uis.no (S.C.S.C.); hildegunn.fandrem@uis.no (H.F.)

**Keywords:** mediation strategies, parenting course, online risks, parent training, youths’ Internet use

## Abstract

The importance of parents in mediating adolescents’ Internet use is documented by many studies in the literature. Very few, however, regard interventions to support parents in this role. We wanted to assess the feasibility and the early outcomes of an Internet-based parent educational group course aimed to support parents in adolescents’ media education. The intervention was conducted with two different groups of parents (N = 20, 75% women; mean age = 46.9 y; SD = 6.3) at different time-points. The intervention included five sessions, during which information about parental mediation strategies was provided and practical exercises based on the Acceptance and Commitment Therapy (ACT) approach were proposed. The intervention generated greater awareness, openness and flexibility and increased parental familiarity with technological devices. Overall, the proposed web-based and group-based parent training model has shown good feasibility and promising early outcomes in supporting parents as Internet mediators.

## 1. Introduction

Improper use of the new forms of communication can affect adolescents’ health, making them more sedentary and more likely to go to bed late, skip meals and eat in front of the computer [1], and it can lead to psychological maladjustment and dependence [2].

Among the most frequent Internet risks there are online aggression, cyber-victimization and aggressive or humiliating messages [3]. Moreover, according to Smahel and colleagues [3], 8–12% of young people have visited sites about how to hurt themselves, ways to commit suicide or to become “very thin”, and up to 17% have viewed hate messages, videos about drugs or violent images. In Italy, 27% of children are in contact with people they have never met face-to-face: in particular, 13–14 year olds (29%) and 15–17 year olds (44%).

Adult mediation of adolescents’ Internet use is therefore relevant. Often, parents are concerned about online risks: the percentage of the negative aspects parents mention when talking about their children and adolescents’ Internet use is more than double than the percentage of the reported positive aspects [4]. Problems related to adolescents’ Internet use have also been found to be a competing factor in poor relationships with parents during adolescence [5].

For these reasons, finding an evidence-based way to encourage parents to think about the Web, its opportunities and its risks, as well as mediation and negotiation strategies and the emotions these generate, could be useful not only for families but also for psychologists and professionals who work with them.

There are several types of parental Internet mediation [5]. Among the best known and most studied are restrictive mediation, which involves impositions and limits on browsing, and active mediation, which involves promoting open discussions about the risks and opportunities of the Web. Both types of mediation seem to limit problematic online behaviors and risks experienced online by the children; however, excessive controls and restrictions can limit children’s opportunities to use the Internet for positive purposes and their digital skills [6]. Moreover, a protective factor in young people’s Internet use appears to be a good relationships with their parents [7].

The lockdown period due to the COVID-19 pandemic has made families even more concerned about their pre-teen and teenage children’s use of the Web, prompting psychologists and educators to find innovative and effective ways to involve parents in their children’s digital media education [7]. 

The vast majority of the literature on web-based parent intervention is related to parents of children or adolescents with specific medical or neurodevelopmental conditions such as obesity [8], cancer [9], diabetes [10], autism [11], asthma [12] or chronic pain [13]. Few studies in the literature have examined the outcomes of web-based and group-based parent training for parents of children or adolescents in the general population. In particular, only one study evaluated an intervention to support parents in their role as mediators of digital content [14]; however, it was specifically concerned with the parental role in the mediation of sexual content. Moreover, most of the online parent trainings discussed in the literature involved watching videos and performing exercises individually by each parent, and almost none involved groups of parents coordinated “live” by an expert.

### 1.1. Effectiveness of Web-Based and Group-Based Parental Interventions

The parent support literature has shown the effectiveness of interventions for parents in terms of increasing parenting success, promoting effective parenting skills and positive parent interactions with their offspring, as well teaching appropriate and functional communication skills [15].

Despite the proven effectiveness of parent training programs, usually delivered face to face (in groups or individually), their positive effects, intensiveness and sustainability are often compromised due to low participation levels and high dropout rates [16]. This could occur due to both practical (e.g., schedules, transportation) and psychological (e.g., family member resistance, beliefs about the treatment process) factors.

For example, in preventive parent training, only 10 to 34% of eligible parents enroll in the course [17]; among those who do enroll, attendance rates are often less than 50% of meetings, and some data show that up to one-third of parents enrolled in the program do not attend even one session [18].

Already in 2009, the Institute of Medicine (IOM) report on the prevention of psychological and behavioral difficulties identified the Internet, media, and other current technologies as appropriate modes of intervention to reach larger segments of the population [19]. Among the advantages of the online mode for this type of intervention are flexibility, convenience, ease of use, consistency of participation (good fidelity rates), reduced time spent traveling for both providers implementing interventions and parents and reduced costs [20]. These benefits eliminate the main factors contributing to low parental participation in traditional models. Moreover, early data on video conferencing for group therapy are promising. For this type of intervention, the literature has reported results similar to those found in face-to-face groups for 125 clients with post-traumatic stress disorder [21] and for clients with traumatic brain injury [22].

The group-based approach offers many advantages in helping normalize challenges faced by parents, opportunities to provide peer support and coping strategies, and the efficiency and cost-effectiveness of the group compared with individual treatment. A review on support group videoconferencing [23] found benefits for the participants, such as engagement with others with similar issues, improved accessibility and the development of knowledge, insights and skills.

### 1.2. The Acceptance and Commitment Therapy (ACT) Approach

The primary goal of many parenting interventions is to teach parents specific skills with which to deal with children and adolescents’ behaviors. Moreover, the impact of restrictions from COVID-19 has been negative for parenting; it has changed parents’ relationships with their offspring by increasing the use of harsh parenting [24]. Given that a good family relationship can be protective against online risks, it is important to support parents and mitigate the negative impact of stress.

To this aim, in addition to general information components with respect to technological devices and hands-on exercises to train parents in the proper use of mediation strategies, psychological exercises were proposed by the psychologist operator [25].

The proposed exercises were based on acceptance and commitment therapy (ACT), a model of reference that has already been shown to be effective in improving one’s adjustment and flexibility in raising children with various difficulties [26]. ACT is a “third-wave” cognitive behavioral intervention aimed at improving our psychological flexibility [27]. Rather than avoiding psychological events, ACT is based on the belief that acceptance and mindfulness are more adaptive responses to life’s inevitability. By experiencing our thoughts, physical sensations and emotions in more flexible ways, acceptance commitment therapists argue that we can reduce the negative behaviors to which they often lead [28].

In the intervention we developed and tested for its formative effectiveness, we built on these components and designed exercises to increase parents’ levels of flexibility, acceptance, and awareness. Walser and Pistorello [29] pointed out that the benefits of offering ACT in groups are many: shared examples in groups can help people normalize their experiences and place problems within the control/avoidance frame; moreover, some of the ACT interventions lend themselves to being physically performed in a group setting, and observing how others show acceptance and willingness can be an encouraging, powerful and challenging experience.

ACT involves a number of exercises focused on its main components [30]: acceptance of thoughts and emotions, even unpleasant ones; and commitment to actions that respect our values. Regarding values, for example, patients in an ACT-based pathway are generally encouraged to think about what they want to do with their lives, not what they do not want to have or feel. This reorientation is achieved by helping patients to reflect on what they want their lives to be and how this is represented in key areas such as family, friends, romantic relationships, leisure, spirituality, health, career, education and community [31].

Moreover, in the parent-training literature, mindfulness sessions have been suggested as a strategy by which to improve parenting skills. These practices may allow parents to use parenting skills more effectively and improve parent–youth interactions, reducing the risk of youth problems [32]. Mindfulness in parenting has been tested with parents of children and adolescents with developmental delays [33], conduct disorders [34], ADHD [35] and special educational needs [36]. In one study [37], parents of children with autism were offered mindfulness sessions in group and online settings. The results showed a reduction in stress compared to the control group that was also maintained during follow-up after three months.

For its characteristics, the ACT approach seems to be promising for interventions aimed at increasing media–parental mediation skills.

Many processes of ACT [27], in fact, can be declined in the context of parental supervision of the Internet:

Acceptance—that is, the willingness to feel emotions without regulating or stopping them—can be translated as “staying with” the frustration at seeing one’s own teenager offspring use the Internet, without intervening immediately.

Defusion—i.e., noting that our thoughts are products of our minds and not absolute realities—can help parents in regulating their own emotions, and this can also be useful in discussing Internet and media use with their adolescents.

Referring to values—the areas of life that are important to us—can remind parents what rules are worth imposing, even in the context of Internet and media use.

Promoting committed action means moving in the valued direction, and it encourages that parents take a step toward the parents they want to be, not just away from what they do not want to be.

Despite these promising premises, to our knowledge, ACT has not been used to date in interventions to improve parenting mediation skills.

### 1.3. Current Study

This study aims to offer an initial assessment of the feasibility and training effectiveness of an online, group-based path developed to support parents in their role as Internet and communication technologies (ICTs) mediators for their adolescent offspring. In the pathway, relevant information about the Internet, Internet risks and technicalities for possible parental mediation (such as the use of software for parental control) was provided, but the core aspect of the pathway was the application of ACT methods to strengthen parents’ skills and ability to reflect and compromise.

In particular, the objectives of the study were as follows: (1) to evaluate the feasibility of an online and group Parent Training intervention to support the management of adolescents’ media use (how accessible, easy-to-participate and engaging the project was); and (2) to examine its formative effectiveness with regards to parent–child communication, application of parental mediation strategies and ways of observing and communicating with their adolescents. Qualitative data were collected and analyzed to understand the utility and feasibility of a web-based delivery model for parents.

Parents of adolescents (since 11 years) were involved as participants in the pathway as adolescence is the critical age when young people increase their use of the Internet, and interactions with peers, also through social media, becomes central in the definition of their identity.

## 2. Materials and Methods

### 2.1. Participants

Parents in our sample were informed about the event through posters shared in psychological and learning centers and schools in northern Italy and shared in parent groups on Facebook. In total, 20 parents, who were split into two groups, answered the call for participating in the course. In the first group, 12 parents (including 1 parental couple) participated (9 mothers and 3 fathers, average age of parents: 43.50, SD: 2.61, range 39–48 years; average age of sons and daughters: 11.58, SD: 1.16, range 11–15 years), while in the second group, 8 parents participated, again including 1 parental couple (6 mothers and 2 fathers, average age of parents: 46.38, SD: 4.96, range 39–54 years; average age of sons and daughters:13.50, DS: 1.41, range 11–15 years). All parents involved in the project were Italian and of Caucasian ethnicity.

The inclusion criteria for admission to the course were (1) being a parent of at least one adolescent aged 11–16; (2) that the son or daughter used the Internet. Parental pairs were also admitted.

### 2.2. Instruments

Qualitative data were collected through semi-structured interviews of about thirty minutes conducted two weeks after the end of the intervention pathways, which were intended to explore two main areas: feasibility of the pathway and outcomes. Questions were, for example: How would you describe your experience? What were your feelings after the first meeting and after subsequent meetings? What did you think worked in the course? How did you find sharing experiences with other parents? How did you find the content of the course? How has your relationship with your son/daughter been these past few weeks? Has there been any change in the way you observe/behave with him/her?

### 2.3. Procedures

The intervention was conducted in two editions with two groups of parents. The first group took place in July 2021 and the second in October and November 2021.

The study obtained the ethical approval of the Catholic University of the Sacred Heart in June 2021. Participants signed the written consent form describing the project and they re-sent the signed form via e-mail.

Parents were also asked to indicate dates and times when it would be easier for them to attend meetings. The majority of parents in both groups indicated evening times (from nine p.m.) as more convenient. The first group preferred a weekly meeting frequency, while the second group preferred a bi-weekly frequency. As a result, the first group of parents went through a 5-week course, while the second group went through a 2 1/2-week course. The intervention was conducted over five weekly meetings, each of 90 min. The platform used was Zoom and the meetings were conducted in the evening hours (9 p.m., a time when all parents were available). The topics of the various sessions are presented in Table 1. Two weeks after the end of the five meetings, parents were asked to participate in individual interviews designed to investigate their impressions of the course and any changes in observation and interaction with their sons or daughters, and they were further asked to complete the post-test questionnaire. These interviews lasted for about thirty minutes each.

The total number of meetings for both groups was five.

The group facilitator was a psychologist experienced in conducting parenting groups and coordinator of the adolescent sector of a psychological service center in northern Italy.

Eleven parents agreed to be interviewed after the five meetings. Qualitative data from the interviews’ transcripts were codified by two independent coders and analyzed using Nvivo software, with a thematic approach [38]. The two coders analyzed parents’ responses separately by highlighting key themes in the interviews and defining codes. After an initial analysis conducted separately, they agreed on what were the main themes that emerged from the interviews.

A combination of an inductive and a theoretical approach was used for the data analyses; we kept previous research as our main theoretical framework, but then adopted an explorative perspective instead of trying to fit the data into pre-existing coding frames. Additionally, a latent rather than a semantic approach was used, which means that we judged the underlying ideas, conceptualizations and assumptions to be more important than finding the surface meanings. Any disagreements between the two coders regarding interpretation of the data were solved via another read through.

As this study was part of a larger research project, quantitative data (self-report questionnaires pre- and post-pathway) were also collected, but the limited sample size of the respondents prevented us from examining them because of the limited possibilities for generalizability (results from the quantitative data are available in the Appendix A).

At the end of the course, all participating parents were emailed a pdf with the topics covered in the course and some suggestions.

**Table 1 healthcare-11-02130-t001:** Topics and exercises of the various sessions.

**Day 1** **Presentations** **Group rules ** **Parenting Values and Goals** **Sharing of difficulties**	Presentations of the group. Sharing of group rules (related to privacy and respect for others’ opinions). Clarification of the differences between parental “values” and “goals”. Exercise 1: Define your values [39]: Parents were asked to identify from a list their three most important values. They were then asked to identify one or more goals to set for themselves as parents that would move in the direction of those values. Descriptions of one’s difficulties in the management of adolescents’ digital devices. Description of a problematic episode related to the management of the Internet by the adolescents expectations about the course. Sharing how they perceive their offspring’s media and Internet use.
**Day 2** **Acceptance** **Empathy and Active listening ** **CBT and Cognitive Distortion**	Discussion about active listening and empathy. Exercise 2: Extreme Video. Parents were shown a video of a device-related argument between a mom and son. Parents have been asked to explain what they themselves felt during the viewing, accepting the emotion without trying to stop it, and to identify with the protagonists of the video, trying to understand the emotions of the characters with different roles.
**Day 3** **Reinforcement and Punishment** **Various applications of Token Economy** **Mindfulness section**	Explanation of the concepts of reinforcement and punishment. Reflections on what constitutes reinforcement for their children. Explanation of the Token Economy strategy (a kind of contract with the child) and its various applications and rules for use. Exercise 3: Create a token economy following the explained rules and share it with other parents. Discussion of what is functional and what can be improved in the various Token economies. Explanation of the usefulness of mindfulness and 5 min mindfulness session.
**Day 4** **Web Risks and Opportunities ** **Cognitive defusion** **Parental Mediation Strategies** **Mindfulness Session**	An explanation of the risks and opportunities of the Internet has been offered. Exercise 4: Metaphors: The use of metaphors to describe problems is a typical ACT exercise [40]. Parents were asked to try to imagine which animal best represented technological devices, which animal best represented their son or daughter, and which animal best represented themselves. The exercise was followed by a brainstorming session on the various representations. Moreover, parents were informed of ways in which to start a conversation about their adolescents’ social network profile and Internet use via two sample videos. Finally, a 5 min mindfulness section.
**Day 5** **Awareness, Flexibility and Compromises** **Describe your parenting through Dixit Cards**	Explanation of the importance of making compromises and not judging yourself, accepting yourself as a parent making mistakes and trying to improve. Exercise 5: Dixit Game Cards. The use of these images for psychological exercises has been found to be functional in the literature for increasing levels of awareness and flexibility [41]. Parents were shown Dixit game cards and asked which one most represented their parenting and why.
**Individual Interviews**	In order to evaluate the functionality of the intervention more thoroughly, individual interviews with each parent who took part in the study were administered at the end of the five meetings. The interviews with parents investigated aspects such a: ease of participation, comments about the time of the sessions and the platform used, what they liked and what could be improved in future courses. Parents were also asked what they felt they had learned from the course and what they felt weak about as parents. Questions posed to parents are reported in Appendix B.

## 3. Results

The participation rate was high in the first group, as only four participants did not attend one of the five meetings (93.3% participation rate in total considering 12 parents and five groups), and a bit lower in the second group (85%), as fix of the eight participants did not attend one of the five meetings. Based on the thematic analysis of the interviews, several areas emerged regarding the feasibility and the outcomes of the parent training intervention. Figure 1 displays a description of the results.

### 3.1. Feasibility

Via the term “feasibility”, we analyzed how easy and smooth parents found participating in the intervention pathway. Based on the analysis of the interviews, the feasibility results were categorized into three themes: general experience, strengths and weaknesses, and technical aspects.

#### 3.1.1. General Experiences

Through the semi-structured interviews, we investigated some aspects related to participation (i.e., how parents experienced the course) and outcomes (i.e., any changes in parents’ thinking or behavior after attending the course). Regarding the overall experience, three themes emerged: engagement; couple; and reflection on technology. In terms of engagement, almost all parents reported that they perceived themselves as engaged and attentive. An important aspect with respect to reported experiences relates to the couple’s setup. Two parental couples (mothers and fathers of the same adolescent) were present in the course (one pair for each edition). In particular, one participant was impressed with her husband’s commitment during the course: “In one of the exercises you made us do, he compared himself to a hippo… and it’s true, he’s very external, I’m much more inside in the education of the children (…). He is very shy, so this amazed me because when he explained to you about the hippopotamus… I was astonished and I said: finally F. has said something…!” One participant who did not attend as a couple also reported sharing course content with her partner: “I socialized with my partner everything I was learning from the lessons from the start to the end, we finished talking at midnight. (…) we also talked about values, what we think we should pay attention to and what our daughter should do independently.” The majority of parents, when asked about the experience in general, also reported that the course made them think about their own ideas about technology. One participant, for example, reported that before the course, “the Internet was all negative”, then he started to think about all the opportunities and became more aware of his reactions. Another participant declared that the course made the parents more aware and that, thanks to the course, parents had the opportunity to learn more about themselves and what they want from their sons and daughters.

#### 3.1.2. Strengths and Weaknesses

Regarding strengths and weaknesses, all parents spoke positively about the pathways being group-based. In particular, four parents spoke of the comparison with other parents as being positive because they felt less alone, understanding that their problems were also those of other parents. Some parents perceived more freedom to express their problems. A mother, for example, reported: “Finally someone says: <Yes, I have this problem!> Here where I live every parent is perfect. I am from a small town…”.

Another participant also reported that the best aspect of the course was that they did not know each other, so they felt more comfortable sharing their problems with their offspring. Someone spoke of an initial awkwardness in sharing their difficulties as a parent, which subsided over the course of the meetings; then, they realized over the course that children’s Internet and social network use was a problem that involved other parents as well. Still with respect to strengths and weaknesses, there were mixed opinions on the practicality of the course: two parents found the exercises practical and easily applicable to real-world settings; a father, for example, reported that he felt like “a part of the whole” and that the course was very hands-on. On the other hand, another two parents found the exercises to be too theoretical. Going into the specifics of the various exercises, the token economy appealed to many parents. A mother stated, for example: “I will definitely implement the token economy. Very smart, not punitive, but rewarding… I was a teacher, so I understand the validity of this technique….”. Two other parents liked the game played through the Dixit cards, in which cards were projected and parents were asked to indicate which represented their parenting style the most. A mother said: “And one part that I also liked was when we played the Dixit game… because to me one of the images… had really struck me, I saw myself in it. And I saw that other parents also chose the same image but interpreted it differently… based on their own perception”. Two parents also enjoyed the exercise in which they were asked to watch an exaggerated video of a parent–child argument scene related to a device and to identify first with the parent and then with the child, describing what the characters might be thinking and feeling. A somewhat cited topic was the cognitive behavioral approach, in which people received information with respect to how thoughts, emotions and behaviors are related. Mindfulness sessions were also appreciated by most parents, and some of them reported benefit from the practice and tried to do it autonomously after the course. One mom said: “Doing mindfulness was also interesting, although a little difficult for me, but I was really immersing myself in the story.” One father disliked the reinforcement-based content and exercises, especially the token economy: “I just didn’t like that on the token economy…you did well to talk about it, so I knew the name, but at the end of the day from that it comes out that one does things because there’s a prize…it’s not like that in a family…you have to do things because you have to do them and that’s it…”.

#### 3.1.3. Technical Aspects

Parents were asked for their opinions regarding the technical aspects of the pathway such as the hour of the sessions, the platform used, the frequency of the sessions and the number of participants. Regarding the time of the meetings, 9 p.m. was considered an appropriate time by the majority of parents, although some reported fatigue at that time of the evening. For none of the parents, the platform used (Zoom) was difficult or problematic. Two of the parents of the second group, who held two meetings a week, said that it would be more effective to propose only one meeting a week. Regarding the number of parents who attended the classes (12 in the first group and 8 in the second), some parents in the first group reported that since this was a project in which a lot of sharing was required, a smaller number of participants might be more adequate. As for the duration of the project, some parents would have liked to have more meetings; however, in general, the majority of parents described the five-meeting commitment as doable: “It’s not that unachievable…once a week at that time…you can arrange it just fine. It’s not a three-four month project. It’s doable!”

### 3.2. Outcomes

In terms of outcomes, five sub-themes emerged from the analysis of the interviews: awareness; flexibility and compromise; increased focus; reflection before action; and familiarization with the technology.

#### 3.2.1. Awareness

The majority of parents reported an increase in awareness with respect to their feelings about their offspring’s device use. Some parents reported that they were able to engage their daughters or sons in this new awareness as well. One mother, for example, reported: “He didn’t realize how much time he was in front of the screen before. And seeing me more prepared, he was the one who came to me one day and said: Mom, I’ve been playing online games all morning! I told him: at least you realize it now!”

#### 3.2.2. Flexibility and Compromises

Another recurring theme was flexibility and compromise. Three parents reported increasing their level of acceptance of their own limitations and the possibility of reaching a compromise. A mother, for example, said: “If he doesn’t follow a rule once in a while I’m more comfortable now because I realized that I also have to survive. Otherwise it would be devastating.” One father reported: “I get that rules are needed, but I can’t alienate them from technology.” Some other parents reported that they had reached compromises through the course in terms of time spent on online games and volume of surfing by their children.

#### 3.2.3. Increased Attention

Parents also reported an increase in attention with respect to their daughter/son’s online activities, checking the rules they gave to their daughters and sons more consistently after the course. Some parents were more prone to know the real online activities conducted by their children: “(…) we want to see if we are really immune to this issue, or if our daughter even in the short time in which she uses the computer is taken by this whirlwind…” Other parents said that after the course, they sought to watch their daughters and sons more carefully and take them out more often.

#### 3.2.4. Reflection before Action

The lesson in which the connections between thoughts, emotions and behaviors were explained received positive feedback: many parents who chose to undergo the interview reported thinking more before taking punitive actions when their children had spent too much time online. One participant stated it this way: “(…) now I have a more conciliatory attitude, more sympathetic, more, let’s say, aimed at making him understand the benefits and the damages, rather than acting as a brake, -Take it off! Disconnect it- I try to make him aware”. Another mother reported: “there are still moments of confrontation, but thanks to the course I was able to reflect, think before I got to the fight with him”.

#### 3.2.5. Familiarization with Technology

Most parents reported increased openness and familiarity with technology. Some reported an increase in time shared with their offspring with respect to what was happening online. Two sentences that exemplify this openness are those of two mothers who participated in the first group: “I had never been interested in these things, but now there has been a small opening on my part, so much so that my child came to me saying <mommy I have to show you a video, you have to see it!> before he never shared anything with me!”; “Now when he calls me and says <mommy come and see this>, I just sit there and watch (…), before I didn’t do it, now thanks to you I’m learning to look out more to this world of his”.

## 4. Discussion

Overall, qualitative results confirmed the formative effectiveness of the course we developed and indicated that a group-based, Internet-based and ACT-based parent training program focused on managing adolescents’ Internet and social network use can be feasible and produce positive outcomes. Parents largely appreciated the group mode of the pathways, and especially the possibility of comparing themselves with peers with similar problems.

An important aspect of the course was that they did not know each other offline, a factor that may have led to greater openness and sincerity in recounting their problems. This result is in line with findings from another study that reported similarly positive feedback for the group mode of parent training for parents whose daughters and sons had ADHD [42].

In particular, the study by Fogler and colleagues [42] found numerous advantages of conducting parenting groups online, such as convenience in terms of the lack of travel expenses or need for a babysitter, and the ability for both parents to be present at the course. Moreover, in our pathway, some parents appreciated the opportunity to participate in the parent training as a couple and stated that they shared all of the topics covered in the course with their partner.

The ability to participate as a couple was also found to be a relevant finding in terms of improved satisfaction with their relationship in another study based on ACT [43]. The same study combined an intervention aimed at improving the techniques of children with disabilities, along with techniques based on ACT, to improve couple bonding, with relevant results in improving parenting stress levels, psychological flexibility, confidence in managing problem behaviors and reduced parenting-related disagreement. This may mean that to work on some problematic or risky behaviors of children, it may be important to involve both parents and work on their relationship with each other and with their child.

Regarding the technical aspects, the participation rate for both editions of the course was high (93.3% for the first groups and 85% for the second). These data, together with the parents’ statements, suggest that the use of technology does not represent a barrier for the parents. The weekly frequency, as opposed to the bi-weekly frequency, was better appreciated. The length of the course was sustainable for all parents, but some would have preferred to run more meetings. The number of participants was also perceived as adequate, although a minority of parents would have preferred fewer participants per group.

This study has limitations that need to be addressed in future research. First, like the feasibility study of Breitenstein and Gross [44], our study involved a small sample of participants. In future work, we plan to validate the pathway conducted in parallel by multiple professionals involving a larger sample, which would also allow us to evaluate the pathway by means of quantitative data and a control sample in order to confirm and clarify our results.

A second limitation pertains to the parents who contacted us to ask about participation. Although they might represent an appropriate population, since these parents were already interested in receiving interventions on Internet and social network mediation, they were already aware of the problematic nature of their situation. To overcome this limitation in future studies, we plan to contact schools and psychological centers and encourage them to offer the course to parents who have not requested it or are unaware of it, but who might benefit from the course.

In addition, our sample was small. The quantitative data we collected could only identify trends of change without providing statistically significant results. The fact that one parent did not like some of the topics proposed is important; it made us realize that, most likely, not all parents will be motivated and committed to a cognitive behavioral theory-based method of learning, and that those who do not may benefit more from other types of interventions. Further investigation should be conducted regarding the characteristics of the parents who are more willing to attend web-based interventions, as well as the characteristics that indicate greater motivation and engagement in preventive interventions.

Despite these limitations, our study provided important information regarding the comprehensive development of web-based intervention or prevention programs for parents.

## 5. Conclusions

In general, qualitative data referring to the parent training course suggests that the course was sustainable for parents, with good fidelity rates and a promising formative effectiveness. Moreover, feedback from parents have been positive.

In light of previous studies which show that parents of children with special educational needs report greater concerns about the virtual world and their children’s the use of the Internet and social networks, this course, and courses with similar characteristics, may be a useful tool for professionals in education, psychologists and psychotherapists who deal with children and adolescents particularly vulnerable to virtual environments and their families. Courses such as this one may also be useful in increasing the involvement of parents in joint activities with schools to prevent cyberbullying, as some previous research indicates that collaboration between schools and parents regarding bullying and cyberbullying might be problematic [45].

Considering the increase in requests for psychological support and the generally unaffordable costs of psychotherapy or individual psychological support, further studies are needed in order to assess the feasibility and effectiveness of online parent interventions. Even from the point of view of sustainability for public or private clinics that provide psychological and educational services, group and web-based courses are an excellent method of reaching as many people as possible while limiting staff costs—and therefore costs for families—and promoting the spread of psychological practices on a wider scale. 

This study also helps highlight the need to implement courses for psychologists on working with groups; such knowledge is increasingly useful in the current world, where more and more families may need psychological support services. The relevance of this parent course and its feasibility and sustainability aspects make it an interesting model by which to develop additional courses on parental involvement in media education, and also those targeting other types of individuals and patients.

## Figures and Tables

**Figure 1 healthcare-11-02130-f001:**
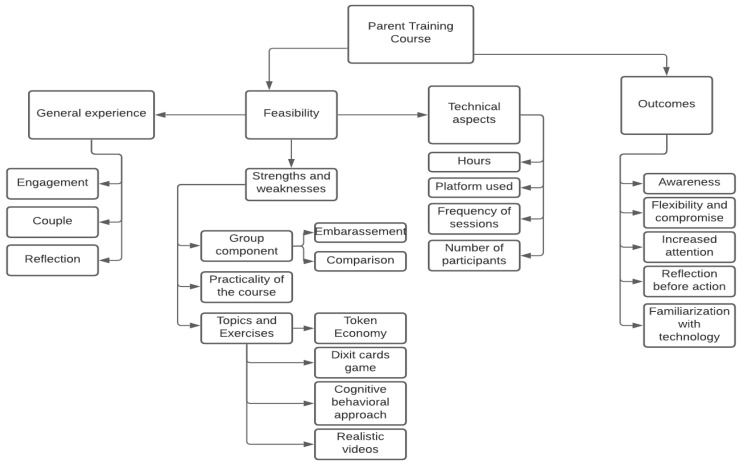
Results description chart.

## Data Availability

The data presented in this study are available on request from the corresponding author. The data are not publicly available due to privacy.

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
