# Peer review of "Feasibility and Early Outcomes of a Parent Training Intervention to Engage Parents in Children’s Media Education"

_healthcare, 2023, doi:10.3390/healthcare11152130_

Round 1

Reviewer 1 Report

Dear authors,

The article focuses on an interesting, relevant, and current topic. However, the data are from an exploratory study of the authors' first approach to parenting education. Thus, the article should be written in this perspective or as a case study. In my opinion, there are some aspects in the article that are not well articulated and presented. More specifically, I mention the following weaknesses:

(a) the number of participants does not allow for generalizations, and statistical analyses also do not allow for the postulated inferences, as the results are not significant and in view of the number of subjects non-parametric statistics should have been used. In addition, are considered both groups of parents simultaneously when they constitute different groups under different intervention conditions. It is mentioned in the abstract but then are grouped in the results section.

b) Authors mention children but parents’ children could be between 11 and 16 years old, which are no longer children and poses other challenges in the development and interaction with children or adolescents.

c) In the introduction authors do not problematize aspects of children's development that parents should also consider related to internet use.

d) on page 3 it is written: "Qualitative and quantitative data were collected in parallel, analyzed separately" but it is not correct because no qualitative data was collected before the sessions.

e) on page 4 is referred the post-test questionnaire but there is no mention when the pre-test took place.

f) The materials and methods section should be organized into participants, instruments, and procedures, to facilitate reading and understanding, as it is not very clear.

g) on page 5, authors mention: "In this way we could compare the formative effectiveness of two levels of intensiveness in the sessions (1 vs. 2 sessions per week)", but this is not clearly considered in the article as they present the results for the two groups together.

h) It is not clear how the categorization of the interviews was made by the two coders and a t-test should not have been used for such a small group of participants, and the two groups should have been separated according to the weeks of intervention.

i) in a case study it could be presented data and graphics discriminated for each of the groups.

j) the discussion needs further development.

l) on page 10 is written: "the quantitative and qualitative data, referring to the parent training course, showed that the course was sustainable for parents, had a good formative effectiveness and was evaluated positively by the parents”, but the results must be interpreted carefully because of what I have already stated above, as the quantitative data cannot be considered in that way.

To sum up, I do not think the article is ready to be published the way it is written. In my opinion, the article should be rejected, since the contribution presented is not properly supported by the results, which are not properly presented.

Author Response

The article focuses on an interesting, relevant, and current topic. However, the data are from an exploratory study of the authors' first approach to parenting education. Thus, the article should be written in this perspective or as a case study. In my opinion, there are some aspects in the article that are not well articulated and presented. More specifically, I mention the following weaknesses:

the number of participants does not allow for generalizations, and statistical analyses also do not allow for the postulated inferences, as the results are not significant and in view of the number of subjects non-parametric statistics should have been used. In addition, are considered both groups of parents simultaneously when they constitute different groups under different intervention conditions. It is mentioned in the abstract but then are grouped in the results section.    

Given that the focus of the article is qualitative data, and the number of participants does not allow for meaningful analysis, we have chosen to remove from the article the analyses performed on the quantitative data.

Authors mention children but parents’ children could be between 11 and 16 years old, which are no longer children and poses other challenges in the development and interaction with children or adolescents.

We used the term “adolescents” to respect the target age group

In the introduction authors do not problematize aspects of children's development that parents should also consider related to internet use.

We have explored these aspects in more detail in the introduction

On page 3 it is written: "Qualitative and quantitative data were collected in parallel, analyzed separately" but it is not correct because no qualitative data was collected before the sessions.

We have corrected this sentence

On page 4 is referred the post-test questionnaire but there is no mention when the pre-test took place.

We removed the quantitative analysis

The materials and methods section should be organized into participants, instruments, and procedures, to facilitate reading and understanding, as it is not very clear.

We have organized the method as suggested

 On page 5, authors mention: "In this way we could compare the formative effectiveness of two levels of intensiveness in the sessions (1 vs. 2 sessions per week)", but this is not clearly considered in the article as they present the results for the two groups together.

The only difference between the two cycles of intervention was the frequency: during the first cycle it was weekly, during the second cycle it was biweekly. The content, exercises and activities of the two groups were the same. We specified this in the text and delete the above sentence

It is not clear how the categorization of the interviews was made by the two coders and a t-test should not have been used for such a small group of participants, and the two groups should have been separated according to the weeks of intervention.

We removed the quantitative analysis

 in a case study it could be presented data and graphics discriminated for each of the groups.

Given the limited sample size, we choose to remove the quantitative analysis and deepen the qualitative results.

The discussion needs further development.

We deepened the discussion by following the reviewers' requests

On page 10 is written: "the quantitative and qualitative data, referring to the parent training course, showed that the course was sustainable for parents, had a good formative effectiveness and was evaluated positively by the parents”, but the results must be interpreted carefully because of what I have already stated above, as the quantitative data cannot be considered in that way.

We have modified the sentence to express more cautiously the results that emerged

Reviewer 2 Report

Thank you for the opportunity to review the manuscript entitled "Feasibility and Early Outcomes of a Parent Training Intervention to Engage Parents in Children's Media Education."  The manuscript describes findings from a study aimed at evaluating an intervention for parents, consisting of five weekly sessions, and targeting parental mediation strategies for their children's internet use.  As such, it could comprise an important contribution to the literature, but there are issues that need to be addressed.  Specific points of feedback are below.

In the first paragraph, the authors describe some "costs" of excessive internet use, such as less exercise.  It is important to incorporate the large body of literature that investigates differential effects of different types of media (i.e., exposure to violence, exposure to harmful messages about body images, etc.).  Media is diverse and its impact varies accordingly.

Based on the description provided, the intervention seems to broadly relate to parents' wellbeing  and parenting skills and strategies (i.e., mindfulness, active listening, reinforcement and punishment, etc.).  These concepts can certainly be applied to parents' interactions with their children regarding internet use, but the intervention seems more general than a training explicitly focused on children's media use.  I think the intervention, and the hypothesized mechanisms for its impact on media supervision, needs to be reframed.  For example, the intervention does not seem to emphasize parents' knowledge about available monitoring software, parents' knowledge about sleep guidelines for adolescents, etc.  Instead, it emphasizes parents' skills for communicating and negotiating with their teens.  These targets, and their hypothesized links to outcomes (i.e., better monitoring of children's internet use) need to be clarified and justified.

There is no information about the facilitator of the groups.  A description of the facilitator, and his or her qualifications, is essential.  

The small sample sizes for the quantitative analyses, and the resultant inability to yield statistically significant findings, is problematic.

Author Response

Thank you for the opportunity to review the manuscript entitled "Feasibility and Early Outcomes of a Parent Training Intervention to Engage Parents in Children's Media Education."  The manuscript describes findings from a study aimed at evaluating an intervention for parents, consisting of five weekly sessions, and targeting parental mediation strategies for their children's internet use.  As such, it could comprise an important contribution to the literature, but there are issues that need to be addressed.  Specific points of feedback are below.

In the first paragraph, the authors describe some "costs" of excessive internet use, such as less exercise. It is important to incorporate the large body of literature that investigates differential effects of different types of media (i.e., exposure to violence, exposure to harmful messages about body images, etc.).  Media is diverse and its impact varies accordingly.

The introduction has been expanded and these aspects have been deepened

Based on the description provided, the intervention seems to broadly relate to parents' wellbeing  and parenting skills and strategies (i.e., mindfulness, active listening, reinforcement and punishment, etc.).  These concepts can certainly be applied to parents' interactions with their children regarding internet use, but the intervention seems more general than a training explicitly focused on children's media use.  I think the intervention, and the hypothesized mechanisms for its impact on media supervision, needs to be reframed.  For example, the intervention does not seem to emphasize parents' knowledge about available monitoring software, parents' knowledge about sleep guidelines for adolescents, etc.  Instead, it emphasizes parents' skills for communicating and negotiating with their teens.  These targets, and their hypothesized links to outcomes (i.e., better monitoring of children's internet use) need to be clarified and justified.

We made it clear in the introduction that the goal of the intervention was to improve parents' mediation and negotiation skills about children and adolescents’ Internet use, given the importance of this kind of communication in both parent-adolescent relationship and safer Internet use.

There is no information about the facilitator of the groups.  A description of the facilitator, and his or her qualifications, is essential.  

We have added the requested information

The small sample sizes for the quantitative analyses, and the resultant inability to yield statistically significant findings, is problematic

We removed the quantitative analysis

Round 2

Reviewer 2 Report

The authors addressed my original comments, but I still do have questions about the connections between ACT & internet supervision, specifically.  Perhaps the authors could add a figure depicting how each component or skill within ACT relates to specific competencies for internet supervision.  And / or the authors could describe how ACT has been applied to other comparable parenting challenges.  These "comparable parenting challenges" should be specific enough to answer the question - does ACT just increase good parenting generally, or how does it specifically target internet supervision?  The authors did address these questions in the most recent version, but additional clarification would be helfpul.

The quality of the English language is fine, although I do recommend another proof-read for grammar and word choice.

Author Response

Dear reviewer,

We have modified the article as you suggested, inserting a clarification (pages 3) of how the components of ACT can be declined to be used for working with parents on mediating Internet use, even though there are currently no other studies in the literature proving its effectiveness for this specific skill. We have reread the article carefully  and fixed some of the English terms.

In addition, as requested in the first reviews, we have deepened the introduction, the method used and the discussion, and included information about the practitioner who conducted the groups and the sample.

Looking forward to your feedback,

The authors